# An Empirical Analysis of Energy Consumption, FDI and High Quality Development Based on Time Series Data of Zhejiang Province

**DOI:** 10.3390/ijerph17093321

**Published:** 2020-05-10

**Authors:** Shaolong Zeng, Yiqun Liu, Junjie Ding, Danlu Xu

**Affiliations:** 1Economics and Management School, Hangzhou Normal University, Hangzhou 311121, China; 2School of Economics, Zhejiang University of Finance & Economics, Hangzhou 310018, China; 3School of Economics, Zhejiang University, Hangzhou 310027, China; junjie_ding@zju.edu.cn; 4Institute of Scientific and Technical Information, Chinese Academy of Tropical Agricultural Science, Haikou 571101, China; danluxu@catas.cn

**Keywords:** energy consumption, FDI, economic growth, empirical analysis, Zhejiang Province

## Abstract

This paper aims to identify the relationship among energy consumption, FDI, and economic development in China from 1993 to 2017, taking Zhejiang as an example. FDI is the main factor of the rapid development of Zhejiang’s open economy, which promotes the development of the economy, but also leads to the growth in energy consumption. Based on the time series data of energy consumption, FDI inflow, and GDP in Zhejiang from 1993 to 2017, we choose the vector auto-regression (VAR) model and try to identify the relationship among energy consumption, FDI, and economic development. The results indicate that there is a long-run equilibrium relationship among them. The FDI inflow promotes energy consumption, and the energy consumption promotes FDI inflow in turn. FDI promotes economic growth indirectly through energy consumption. Therefore, improving the quality of FDI and energy efficiency has become an inevitable choice to achieve the transition of Zhejiang’s economy from high speed growth to high quality growth.

## 1. Introduction

Since the start of the 40 years of the period of reform and the policy of opening up, China has persisted in making active use of foreign capital. China’s FDI flow and the volume of GDP have risen dramatically. At the same time, energy consumption has seen a significant increase. China has become the world’s largest energy consumer, accounting for 23.2% of global energy consumption and 33.6% of global energy consumption growth. The traditional economic development remains highly energy-intensive, with high energy consumption, high pollution, and high emissions. With rapid economic growth, the inefficient use of resources and increasing environmental pollution have caused extensive damage to the environment. To solve this problem, China needs to shift to high quality and sustainable growth, with a high-end industrial chain, high quality economic development, and efficient operation transformation. “Clear waters and green mountains are as good as mountains of gold and silver.” President Xi Jinping made this dialectic notion in his speech while he held office in Zhejiang. In China’s 13th Five-Year Plan, the Chinese government aims to reduce energy intensity by a total of 15% from 2016 to 2020. Chinese Premier Li Keqiang reported that China’s energy intensity fell by 5% in 2016 at China’s National People’s Congress. Therefore, exploring the relationship among economic growth, energy consumption, and Foreign Direct Investment (FDI) will help us to understand how the China’s economy shifts from the old equilibrium to the new equilibrium.

FDI is an important factor for rapid economic growth in China. Since the early 1990s, China has become one of the largest recipients of FDI among the developing countries. Neoclassical growth theory outlines the three factors necessary for a growing economy. These are labor, capital, and technology. The relationship between capital and labor of an economy determines its output. Technology is thought to augment labor productivity and increase the output capabilities of labor. New institutional economics incorporates the theory of institutions—laws, rules, customs, and norms—into economics. It builds on, modifies, and extends neoclassical theory. It believes the institutions play an important role in economic development. FDI inflow can promote economic development for a developing region or country. First, FDI improves the overall productivity. When FDI flows in, capital accumulates in the industrial sector, city income grows, and surplus labor from rural areas pours in. This movement improves developing country employment scales and structures and leads to productivity improvements. Second, FDI can enhance technology transfers to domestic firms through spillovers. FDI often comes with new technologies and innovations. It is potentially an important source of productivity growth as it may help host country domestic industries catch up with the international technology frontier and encourage the host country to optimize and upgrade its industrial structure. Furthermore, FDI helps the host country to establish a unified, open, orderly, and competitive market system. In order to attract FDI, government authorities must fully consider the impact of their policies and measures on market competition, creating a transparent, fair competition environment. A fair and open investment environment, in turn, attracts more FDI inflow. Yue et al. (2016) reckoned that FDI can not only affect the economic growth, but also affect the environmental protection of the host country [1]. When evaluating the impact of FDI on China’s economy, we should not just evaluate the impact of FDI on economic growth, but also consider the impact of FDI on energy and the environment.

There are numerous research works on FDI that seek to investigate its impact on economic growth. However, few have been done on the relationship among energy consumption, FDI, and economic development. Empirical studies on examining the dynamic processes among them are even fewer. Zhejiang, located on the eastern coast of China, is a pioneer in China’s reform and opening up. A high-end equipment manufacturing industry is one of the pillar industries and continues to grow at a rapid pace in Zhejiang. The demand for energy remains high. The study on the relationships among energy consumption, FDI, and economic growth in Zhejiang Province can not only provide a theoretical framework for Zhejiang’s economic transformation and development, but also provide a good paragon of high quality development for other provinces in China. The VAR model is used to determine the relationships of energy consumption, FDI, and economic growth in Zhejiang. Cointegration tests, impulse responses, and variance decomposition are used to test the results.

The rest of paper is organized as follows: Section 2 provides a review of the literature. Section 3 contains a briefly discussion on economic growth, energy consumption, and FDI in Zhejiang. The theoretical analysis, description of data, and summary statistics are discussed in Section 4. Empirical results and the robustness test are presented in Section 5 and Section 6, respectively. Section 7 contains conclusion and policy recommendations.

## 2. Literature Review

The relationship among energy consumption, FDI, and economic growth is a hot topic in economics. The existing research mainly focuses on the following three perspectives: energy consumption and FDI, energy consumption and economic growth, and FDI and economic growth.

### 2.1. Energy Consumption and FDI

Sun et al. (2011) conducted an empirical test on panel data of 74 high income, middle income, and low income countries from 1985 to 2008. They found that energy consumption intensity decreased slightly as FDI increased [2]. Zhao et al. (2007) used index decomposition analysis (IDA) to analyze changes in energy consumption over time, arguing that there was a strong relationship between industrial distribution of FDI and energy consumption in China. This implied that industrial distribution of FDI has played an important role in Chinese energy consumption [3]. Li and Qi (2016) used the industrial panel data of each province and applied the Two-Stage Least Squares (TSLS) and GMM methods, finding that FDI has a negative effect on China’s industrial energy consumption [4]. Zhang et al. (2017) obtained a similar result [5]. Latief and Lefen (2019) analyzed the causality among the FDI in the power and energy sector, the energy consumption, and the economic growth of Pakistan in 1990–2017. The sector-wise flow of FDI revealed that the power and energy sector of Pakistan had received a higher amount of FDI than other sectors of the economy with trends of energy production and energy usage [6]. Zhang and Xu (2016) reported that FDI and energy consumption structures were beneficial in improving carbon productivity in resource-intensive sectors [7].

### 2.2. Energy Consumption and Economic Growth

Many studies tried to identify the causal relationship between energy consumption and economic growth. Yin and Wang (2011) used the Engle-Granger two step method, cointegration analysis, and the Granger causality test to identify the relationship between China’s energy consumption and economic growth based on the data for the period of 1953–2008. The results showed that although some short run fluctuation existed, there was a stable equilibrium in the long run. A single direction causality from energy consumption to economic growth was found [8]. The cointegration model and Granger causality test were used to analyze the relationship between the energy consumption and the export in Guangdong (Shi, 2008). They found that there was a positive relationship between energy consumption and export. Energy consumption was the Granger cause of export, while export was not the Granger cause of energy consumption. The relationship was steady and had no structural change. This indicated that export highly depended on energy consumption in Guangdong [9]. Latief and Lefen (2019) confirmed the presence of a positive bi-directional causal relationship between energy consumption and economic growth through empirical research [6].

However, Ma et al. (2012) investigated the causal relationship between energy consumption and economic growth for China from 1978–2008, using a multivariate framework based on the neo-classical production function where economic output, energy consumption, capital stock, and labor level were treated as variables. They applied the vector error correction model (VECM) and the Johansen-Juselius cointegration test to examine the existence of cointegration and Granger causality in the long and short run. Their results suggested there was no Granger causality between energy consumption and economic growth in the short run, but a unidirectional causality running from economic growth to energy consumption in the long run [10]. Werner K.R., et al. (2018) used panel data to identify the causality relationship between the GDP and energy consumption for 74 countries from 1972 to 2014. A bi-causality between electrical energy consumption and GDP was found in the long run, while single direction causality was verified from GDP to electricity consumption in the short run [11].

### 2.3. FDI and Economic Growth

Regarding the relationship between FDI and economic growth, scholars generally believe that technological spillovers brought by FDI play a positive role in economic growth.

Cheng (2002) analyzed the correlation between FDI and open economic growth in three different income countries in the context of new growth theory. She set up the model of foreign direct investment-led open growth and tested the effect of FDI on economic growth in a cross-country regression framework. The results of 65 countries showed that the growth enhancing effects of FDI were more significant in high income countries than those of middle and low income. In addition, FDI had positive effects on increasing total factor productivity and economic growth in China, which linked the scale of FDI inward and human capital [12]. Liu and Li (2013) concluded that there was a two way positive correlation between FDI and economic growth in China based on the VAR model. According to the 1985–2010 time series data, the VAR model including three variables—FDI, economic growth, and employment—were applied, showing that the FDI played a significant role in promoting China’s economic growth and employment; China’s economic growth had a positive impact on FDI inflows, but it was not stable. There was an interactive promoting relationship between FDI, economic growth, and employment [13]. Chen and Li (2011) conducted an empirical analysis of data in Shandong Province and found that there was a long-term equilibrium relationship between FDI and economic growth in Shandong Province and that FDI was the Granger cause of economic growth [14]. Comes, C.-A., et al. (2018) investigated the impact of FDI and remittances on economic growth, using panel data of seven countries from Central and Eastern Europe with a gross domestic product (GDP) per capita under 25,000 $. They found a positive impact of both FDI and remittances on GDP, but the influence of FDI was higher in all analyzed states, with accepting the assumption of ceteris paribus principles in limiting research caused by other possible determinants [15]. Yue, S., et al. (2016), Wang, C.-N., et al. (2019), Dkhili and Dhiab (2018), Zhu and Ye (2018), and Kurecic and Kokotovic (2017) drew a similar conclusion on FDI and economic growth [1,16,17,18,19].

However, some researchers had different perspectives. He and Wang (2011) used cointegration theory and the error correction model to reassess the relationship between economic growth and FDI involving BRIC countries. The empirical analysis found meaningful Granger causality relationships between FDI and the growth of Brazil, Russia, and India in the long term, but no long-term Granger causality relationship existed in China. In the short term, FDI did not exert a robust, positive influence on growth except in Russian, while economic growth did attract inward FDI except in India [20]. Gherghina, Ș.C., et al. (2019) used panel data for 11 Central and Eastern European countries from 2003 to 2016, and the quantitative outcomes by means of panel data regression models led us to conclude that a non-linear association occurred between FDI and GDP per capita [21]. Belloumi and Alshehry (2018) investigated the causal links between domestic capital investment, FDI, and economic growth in Saudi Arabia over the period 1970–2015 by using an Autoregressive Distributed Lag (ARDL) bounds testing to cointegration approach. The results showed that in the long term, there was negative bidirectional causality between non-oil GDP growth and FDI [22]. Dinh, T. -H., et al. (2019) examined and provided additional and relevant quantitative evidence on the impact of FDI on economic growth, both in the short run and the long run in developing countries of the lower middle income group in 2000–2014. The results of this study showed that FDI helped stimulate economic growth in the long run, although it had a negative impact in the short run for the countries in this study [23].

### 2.4. Causality between Energy Consumption, FDI, and Economic Growth

There were some literature works discussing energy consumption, FDI, and economic growth together. Huo (2015), based on China’s provincial panel data from 2001 to 2010, studied the relationship among economic growth, FDI, and energy consumption using simultaneous equation models. The empirical research results showed that for the whole country, a bidirectional causal relationship between economic growth and FDI was found. There was a unidirectional causal relationship from FDI to energy consumption. For the high GDP per capita provinces, a bidirectional causal relationship between economic growth and FDI was found. There were unidirectional causal relationships from economic growth and FDI to energy consumption. For the middle GDP per capita provinces, there were bidirectional causal relationships between economic growth and FDI and economic growth and energy consumption, and there was a unidirectional causal relationship from FDI to energy consumption. For the low GDP per capita provinces, a bidirectional causal relationship between economic growth and FDI was found. There were unidirectional causal relationships from economic growth and FDI to energy consumption [24]. Hussain and Nor (2011) examined the causal relationship between electricity consumption, Consumer Price Index (CPI), GDP, and FDI with time series data used for these variables for the 1971 to 2009 period in Malaysia. VECM was employed to estimate the causal relationship. The result for long run causality from electricity consumption to FDI, GDP growth, and inflation was found to be significant. The economic growth in turn stimulated the growth of power consumption, and the increase in FDI would help reduce power consumption and promote economic growth [25].

### 2.5. FDI and Sustainable Development

FDI is important for sustainable development. There are three major hypotheses: the FDI halo hypothesis, the pollution havens hypothesis, and the environmental Kuznets curve. Ma, C.-Q., et al. (2019) found that FDI had a significant positive effect on the upper emission provinces of the manufacturing industry, which proved that there was a pollution paradise hypothesis in China’s manufacturing industry, but no halo effect hypothesis [26]. Liu and Kim (2018) found the pollution havens hypothesis to be true for FDI and GDP among Belt & Road Initiative (BRI) countries, after adopting weight values in the Panel Vector Autoregression (PVAR) estimation [27]. Jun, W., et al. (2018) examined the effect of FDI on pollution in China, for the period 1982 to 2016. The results indicated that FDI created pollution havens in China [28].

There are different views. Yan and Hu (2020) argued that an increase in the FDI in a city was associated with a decline in SO_2_ emissions in the same city, indicating that the pollution haven hypothesis did not hold in eastern China. However, the spillover effect of FDI was positive, indicating that a larger FDI in neighboring cities tended to raise SO_2_ emissions in the local city [29]. Li, Z., et al. (2019) investigated the impact of FDI on environmental performance by constructing a panel quantile regression model. They found that FDI had an insignificant influence on EP (Environmental Performance), and the impact of FDI on EP between developed and developing countries was heterogeneous. Furthermore, there was heterogeneity regarding the effect of FDI on EP at different quantiles of EP in developed countries [30].

Hu, J., et al. (2019) investigated the environmental regulatory threshold effect of labor-based FDI and capital-based FDI in terms of their green technology spillover. The results showed that environmental regulation was unable to promote the green technology spillover of labor-based FDI significantly. However, intensifying environmental regulation could reduce the negative effect of labor-based FDI on green total factor productivity. The effect of environmental regulation on the green technology spillover of capital-based FDI is more complex. In the static linear model, environmental regulation could significantly promote the green technology spillover of capital-based FDI [31]. Wang, Y., et al. (2019) found that China’s two way FDI coordinated development presented a significant braking effect on carbon emissions during the research period [32]. Adamu, T.M., et al. (2019) revealed that energy consumption, export variety, FDI, and income positively contributed to environmental degradation in India [33].

### 2.6. Summary

Energy consumption, FDI, and economic growth are dynamic processes that affect and optimize each other in the actual economic development process. For the development of Zhejiang’s economy, the process of marketization can especially reflect the close relationship among them. Therefore, this paper uses the VAR model to explore the interrelationships of energy consumption, FDI, and economic growth in Zhejiang Province through cointegration tests, impulse responses, and variance decomposition.

## 3. Economic Growth, Energy Consumption, and FDI in Zhejiang

### 3.1. Economic Growth and Energy Consumption

Economic growth relies on the increase of energy consumption. Especially in the early stages of economic growth, both energy consumption and economic development show upward trends. Figure 1 shows the energy consumption and GDP of Zhejiang.

From 1993 to 2017, the average annual growth rate of energy consumption in Zhejiang was 6.82%, and the average annual growth rate of GDP was 14.07%. In 1993–2001, energy consumption and GDP grew slowly and steadily. In 2002–2010, the growth rate of energy consumption and GDP increased significantly. Energy consumption and GDP grew by 9.95% and 14.93% annually, respectively. After China’s accession to the WTO, Zhejiang promoted a higher level of opening up. The substantial increase in international trade boosted the energy consumption and GDP. In 2011–2017, the growth rate of energy consumption slowed down gradually, with an average annual growth rate of only 2.39%. The average annual growth rate of GDP was slightly higher than energy consumption.

### 3.2. Energy Utilization Efficiency

Compared to the absolute value of energy consumption, energy efficiency measured by energy consumption intensity is a more accurate estimate of energy consumption. The greater the intensity, the lower the energy utilization efficiency. Figure 2 shows the trend of energy consumption intensity in Zhejiang from 1993 to 2017.

The regression line of energy consumption intensity in 1993–2017 is shown in Figure 2, with R^2^ = 0.8208. It shows a decreasing trend of energy consumption intensity in Zhejiang, which implies that Zhejiang’s overall energy efficiency has been improving continuously. However, if compared with other developed provinces in China, there is still much room for improvement in Zhejiang’s energy efficiency. In 2015, the energy consumption intensity in Zhejiang (0.46) not only lagged behind Beijing (0.338) and Shanghai (0.45), the two most developed municipalities in China, but also lagged behind Guangdong (0.35) and Jiangsu (0.43), which had a similar industrial structure and level of economic development.

### 3.3. FDI

#### 3.3.1. FDI Growth

FDI plays an important role in contributing to economic growth in China. Figure 3 shows the actual FDI in Zhejiang from 1993 to 2017. China’s accession to the World Trade Organization (WTO) in 2001 was the turning point. Before entering the WTO, the scale of FDI in Zhejiang fluctuated slightly and did not show an obvious change. FDI in Zhejiang had a rapid growth after 2001, except the shock caused by the global financial crisis of 2008/2009. The average growth rate of FDI in Zhejiang was approximately 15.78% in 2001–2017. It is critical to note that the growth rate of FDI in Zhejiang Province slowed down in 2016–2018, Zhejiang’s FDI encountered a bottleneck under the traditional economic development pattern. Its economy has undergone a significant structural transformation.

#### 3.3.2. Changes in the Industries Structure of FDI

In Zhejiang, the industries that received the most FDI inflow were secondary and tertiary industries (see Table 1). The proportion of primary industry was less than 1%. This was consistent with the characteristics of Zhejiang, a major export processing zone in China’s foreign trade. Since 2013, the proportion of FDI in the tertiary industry surpassed that of the secondary industry, and the gap between secondary and tertiary continued to widen. In the past ten years, the proportion of FDI in the secondary industry fell by half, while the proportion in the tertiary industry doubled. This implied that the tertiary industry became the main industry that attracted FDI in Zhejiang. The proportion of tertiary industry in FDI increased significantly from 30.33% in 2008 to 65.25% in 2017.

The distribution and allocation of FDI inflow among the sectors is shown in Table 2. As we can see from Table 2, FDI inflow in the secondary industry was mainly concentrated in manufacturing. The share of manufacturing in FDI (S0) declined from 67.8% in 2008 to 30.52% in 2017. In the manufacturing sector, the proportion of the four sub-sectors in FDI showed a decreasing trend. In 2017, electronic equipment manufacturing (S2) took the largest share (3.81%); chemical products’ manufacturing (S1) accounted for 3.67%; general equipment manufacturing (S4) and special equipment manufacturing (S3) accounted for 2.38% and 1.37%, respectively. In 2008–2017, the proportion of sub-sectors of the tertiary industry in FDI increased except real estate. Wholesale and retail (T2) trades had the most significant change, which increased by 10.99 percentage points. Scientific research (T5) increased by 10.49 percentage points. It is noteworthy that the financial intermediation sector (T6) increased significantly, from 0.03% to 3.23%.

## 4. Empirical Methodology and Data

### 4.1. Establishment of VAR Model

Vector auto-regression (VAR) is a model based on the statistical properties of the data. The VAR model constructs the model by using each endogenous variable in the system as a function of the lag values of all endogenous variables in the system; thereby extending the univariate autoregressive model to a “vector” autoregressive model composed of multiple time series variables. The VAR model can predict interconnected time series systems and analyze the dynamic impact of random disturbances on variable systems, thereby explaining the impact of various economic shocks on the formation of economic variables. The mathematical expression of the VAR (p) model is:*y_t_* = *A*_1_*y*_*t*−1_ + *A*_2_*y*_*t*−2_ + … *A_p_y*_*t*−*p*_ + *Qx_t_* + *e_t_*  *t* = 1, 2, …, *T*(1)
where *y_t_* is an endogenous column vector of order 3 × 1, *e_t_* is an error column vector of order 3 × 1, *x_t_* is an exogenous column vector of order K × 1, *A*_1_, *A*_2_ ⋯ *A_p_* are 3 × 3 order matrices, and 3 × K order matrices *Q* are the coefficient matrices to be estimated.

### 4.2. Data and Definitions of the Key Variables Used

The annual data of FDI inflow, economic growth (GDP), and energy consumption (ENC) in Zhejiang Province from 1993 to 2017 were used. FDI inflow is the total flow inward of foreign investment actually utilized. All data were obtained from Zhejiang Statistical Yearbook. In order to eliminate the impact of exchange rate fluctuations, we converted GDP to U.S. dollars based on the average annual exchange rate during 1993–2017 based on data from National Bureau of Statistics. To eliminate the impact of price changes, GDP and FDI were adjusted by CPI, with 1993 as the base period. In order to eliminate heteroscedasticity, this paper applied natural log transformation on the condition of keeping the cointegration relationship of the original data. After taking the logarithms, the three variables were transformed to LNFDI, LNGDP, and LNENC. EVIEWS8.0 was used for the empirical analysis. Summary statistics are shown in Table 3.

The correlations of the variables are shown in Table 4, indicating energy consumption to be positively correlated with FDI inflow and real GDP.

## 5. Empirical Results

### 5.1. Stationarity Check

We chose the VAR model to identify the relationships between energy consumption, FDI, and economic growth in Zhejiang by using cointegration tests, impulse responses, and variance decomposition.

#### 5.1.1. Augmented Dickey-Fuller Unit Root Test

Due to the potential hysteresis of the time series data, which was likely to influence the accuracy of the results, the augmented Dickey-Fuller (ADF) unit root test was conducted to check the stationary property of each variable. All variables were integrated of order one at the 10% significance level, as shown in Table 5. Therefore, cointegration tests could be performed to determine whether there was a long-term equilibrium relationship between variables.

#### 5.1.2. VAR Lag Order Check and Cointegration Test

It was necessary to check the lag order by constructing a VAR model. Table 6 shows the lag order of VAR under five different criteria, which indicated that the lag order of the VAR model was three.

VAR was unconstrained, while cointegration was constrained. The optimal lag order of the cointegration test was the optimal lag order of VAR minus one. As the lag order of the VAR model shown in Table 6 was three, the lag order of the Johansen cointegration test was two. The results of the cointegration test are shown in Table 7. According to Table 7, the trace test and max-eigenvalue test indicated one cointegrating equation(s) at the 0.05 level.

### 5.2. VAR Estimate

The results of VAR estimation are shown in Table 8.

According to the parameter estimation results in Table 8, the regression equation can be written as follows:DLNENC = 0.0308 − 0.3523 × DLNENC(−1) + 0.0297 × DLNENC(−2) + 0.3000 × DLNFDI(−1) + 0.0739 × DLNFDI(−2) + 0.2874 × DLNGDP(−1) − 0.0782 × DLNGDP(−2)(2)
DLNFDI = 0.0643 − 1.4844 × DLNENC(−1) − 0.3837 × DLNENC(−2) + 0.7471 × DLNFDI(−1) + 0.3950 × DLNFDI(−2) + 0.8138 × DLNGDP(−1) − 0.3617 × DLNGDP(−2)(3)
DLNGDP = 0.0160 + 0.1748 × DLNENC(−1) + 0.3615 × DLNENC(−2) + 0.0911 × DLNFDI(−1) − 0.1766 × DLNFDI(−2) + 0.4314 × DLNGDP(−1) + 0.1376 × DLNGDP(−2)(4)

## 6. Robustness Test

In order to identify the dynamic relationship among energy consumption, FDI, and economic growth in Zhejiang more accurately, we performed a VAR stability check and Granger causality test. Then, we constructed the impulse response and variance decomposition.

### 6.1. AR Roots Test

The test results are shown in Table 9 and Figure 4. We confirmed that no root lied outside the unit circle, as can be seen in Figure 4, meaning that the VAR models conformed to the stability condition.

### 6.2. Granger Causality Test

The results of the Granger causality test are presented in Table 10. The results indicated that only DLNFDI Granger caused on DLNENC and DLNGDP at the 0.1 level, which was consistent with the findings of Chen and Li (2011).

### 6.3. Impulse Response

The impulse response is the reaction of any dynamic system in response to some external change. As the VAR model conformed to the stability condition, impulse response analysis was used to examine the current and future impact of a single disturbance on all endogenous variables. With one standard deviation impact on the disturbance term, we obtained the impact on the current and future values of the endogenous variables. The results from the Cholesky impulse function are shown in Figure 5, Figure 6 and Figure 7.

Ten periods were used for the lag interval of the impulse response in this paper. It can be seen from the response curve that the responses all tended to zero, indicating that the VAR model was stationary. The impulse response of energy consumption to one standard deviation is shown in Figure 5. If we applied a positive shock to FDI and GDP in the current period, the response of energy consumption was highly positive, with a slightly declining trend in the long term. It is possible to observe that the positive effect of FDI on energy consumption was greater than the positive effect of GDP on energy consumption. After a lag period of FDI inflows, such as building plant production lines and obtaining production and operation licenses, a short period of time was needed to restart production. A large part of FDI was concentrated in high energy-consuming industries. At the same time, the economic development of Zhejiang could also encourage industries to scale up, which would lead to an increase in energy consumption.

The impulse response of FDI under the Cholesky one standard deviation is shown in Figure 6. The impact of energy consumption on FDI reached its maximum in the first period and declined gradually. The impact was close or equal to zero after six periods. The cumulative response of FDI to energy consumption was 0.1778, which indicated that the increase in energy consumption in Zhejiang stimulated FDI inflow for a long time. The impact of GDP on FDI lasted four periods, with very little impact in the fifth period. The impact of GDP on FDI was positive and significant. With the increasing GDP, FDI inflow increased in Zhejiang.

Figure 7 shows the impulse response paths of DLNGDP to DLNEC and DLNFDI under the Cholesky one standard deviation. The impact of FDI on GDP fluctuated in the previous periods, then gradually eased after the fourth period, and the overall value was low. This indicated that FDI impact on GDP was weak and complex. If we applied a positive shock to energy consumption, the response of GDP was significantly positive, with a decreasing trend. To some extent, the economic growth of Zhejiang still depended on the increase in the scale of energy consumption.

### 6.4. Variance Decomposition

Variance decomposition was used to further evaluate the importance of different structural shocks by analyzing the contribution of each structural shock to changes in endogenous variables, which are usually measured by variance. Therefore, variance decomposition could describe the relative importance of different shocks to FDI, GDP, and ENC. Based on the VAR model above, we applied variance decomposition. The results are shown in Table 11.

The fluctuation of FDI was mainly explained by its own impact and the impact of energy consumption. The impact of GDP on FDI was minimal (See Table 11). In the initial period, the impact of energy consumption was slightly greater than the impact of FDI itself. However, the impact of energy consumption gradually weakened, while the impact of FDI itself gradually strengthened. The impact of FDI shocks stabilized at about 58.5%, indicating that FDI still had an important role in Zhejiang’s opening economy.

For the fluctuation of energy consumption, the results of variance decomposition showed that only energy consumption had an impact itself in the early period. Although the impact gradually declined, it remained at a high level at 63.4%, and the impact of FDI could explain 34.7%. Therefore, there was a two way strong relationship between energy consumption and FDI.

The variance decomposition of GDP showed that the impact of energy consumption was also very significant. While the impact of GDP’s own shock was more significant in the initial stage, it gradually weakened over time. The proportion of FDI impact on GDP and the proportion of GDP impact on FDI were relatively small, indicating that the interaction between them was weak.

## 7. Conclusions and Policy Implications

### 7.1. Conclusions

This study examined the energy consumption, FDI, and high quality development in China by examining Zhejiang from 1993 to 2017. The results of cointegration tests showed that there was an equilibrium relationship among Zhejiang’s energy consumption, FDI, and economic growth in the long run. Impulse response analysis and variance decomposition showed that, in terms of short-term dynamic relationship, there was a strong positive relationship between FDI and energy consumption. This was consistent with the findings of Zhao et al. (2007) and Latief and Lefen (2019). In turn, energy consumption stimulated the increase of FDI. Although there was a time-lag between energy consumption and FDI, the effect was strong and persistent.

On the other hand, energy consumption had a strong positive impact on economic growth, which was consistent with the findings of Yin and Wang (2011) and Shi (2008). After China’s accession to the WTO, a higher level of opening up boosted the energy consumption in Zhejiang. Taking the advantage of convenient transportation, Zhejiang took the lead in entering high energy-consuming industries. These high energy-consuming industries played a very important role in promoting economic growth in Zhejiang. Then, the increase in energy consumption attracted more foreign investment in Zhejiang. Therefore, a closed-loop chain of “FDI-energy consumption-economic growth” was formed.

### 7.2. Policy Implications

Zhejiang boasts a developed private economy and advances the construction of industry cluster areas and economic development zones for large enterprises and manufacturing. With the economic expansion, the marginal benefits of energy consumption decrease continuously, and the traditional economic growth method is difficult to sustain. In order to make full use of foreign investment, reduce energy consumption intensity, and promote stable economic growth, Zhejiang needs to pursue economic development in an environmentally sustainable manner and foster green growth of the economy. Based on this, we propose the following policy recommendations.

(1) Coordinate the relationship among energy consumption, economic growth, and FDI to achieve high quality economic development: The results of the cointegration test showed that there was a long-run equilibrium relationship. Therefore, from the perspective of long-run stable growth, policy makers should strike a balance among them, shifting from the old equilibrium to the new equilibrium, and achieve high quality economic development.

(2) Optimizing FDI structure and promoting industrial upgrading: It has been proven that the increase in FDI played an important role in promoting industrial development and economic growth in Zhejiang province. Zhejiang should improve the level of opening up and the quality of investment by encourage the development of low energy consumption and high output industries, as well as attract high quality foreign capital inflows. It is necessary to guide the foreign capital to industries with lower energy consumption. For example, taking advantage of agriculture and fishing resources in Anji and Zhoushan of Zhejiang, the government could introduce advanced machinery, equipment, and management technology through FDI to increase productivity and reduce energy consumption. The government could also attract FDI by making use of talent advantages in the fields of finance and the Internet to develop the tertiary industry in Zhejiang.

(3) Improve the environment for FDI and enhance technology spillover effects of FDI: The development of private enterprises plays a decisive role in promoting Zhejiang’s economic growth and improving people’s living standards. However, private enterprises have disadvantages such as outdated industrial structure and low management level. The technology spillover effects of FDI can solve these problems. However, in reality, foreign capital is only investing in downstream production and processing cycles in Zhejiang, but not setting up R&D institutions. Therefore, the technology spillover effect of FDI in Zhejiang is not obvious, and its driving effect on the private economy is weak. It is necessary to introduce a comprehensive protection system of property rights and copyrights, improve the policies of technology incentives, and support the guidance of FDI in the concentration of high-tech industries, as well as promote the joint establishment of mixed ownership enterprises by private capital and foreign capital.

(4) Develop the digital economy, reduce energy consumption intensity, and improve energy efficiency: Zhejiang has gathered a number of world-leading digital enterprises, such as Alibaba. Zhejiang has a first-mover advantage in the development of the digital economy. The construction of the “Internet +” and “AI +” (artificial intelligence) can promote traditional industrial digitalization. By supporting manufacturing companies in using big data, AI and other advanced technologies can speed up technological transformation and equipment upgrades. Production efficiency can also be improved by reducing energy consumption intensity and optimizing energy efficiency. Therefore, high quality development can be achieved.

## Figures and Tables

**Figure 1 ijerph-17-03321-f001:**
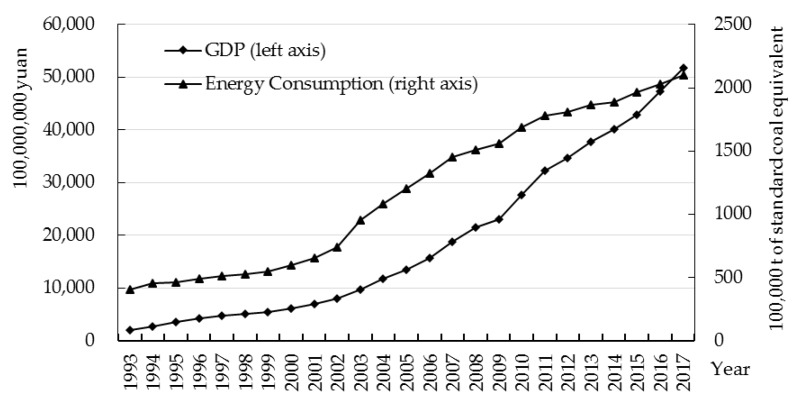
Energy consumption and economic growth in Zhejiang Province (1993 to 2017). Data source: Zhejiang Statistical Yearbook.

**Figure 2 ijerph-17-03321-f002:**
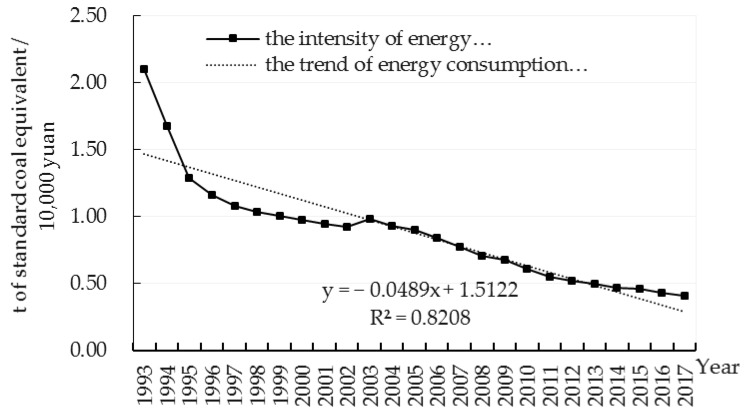
Energy consumption intensity in Zhejiang Province (1993 to 2017). Data source: China Environmental Statistics Yearbook, Zhejiang Statistical Yearbook.

**Figure 3 ijerph-17-03321-f003:**
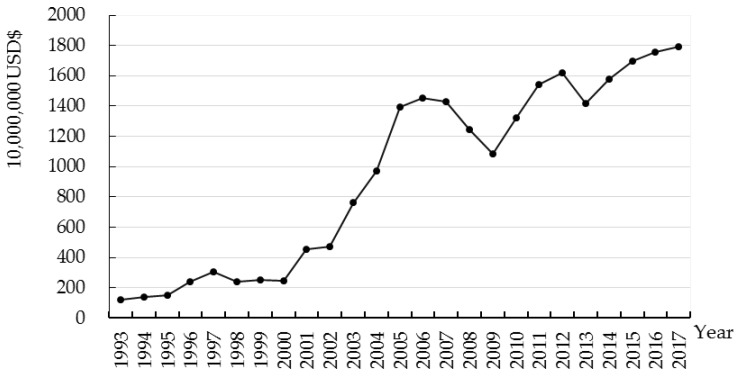
FDI in Zhejiang Province (1993 to 2017). Data source: Zhejiang Statistical Yearbook.

**Figure 4 ijerph-17-03321-f004:**
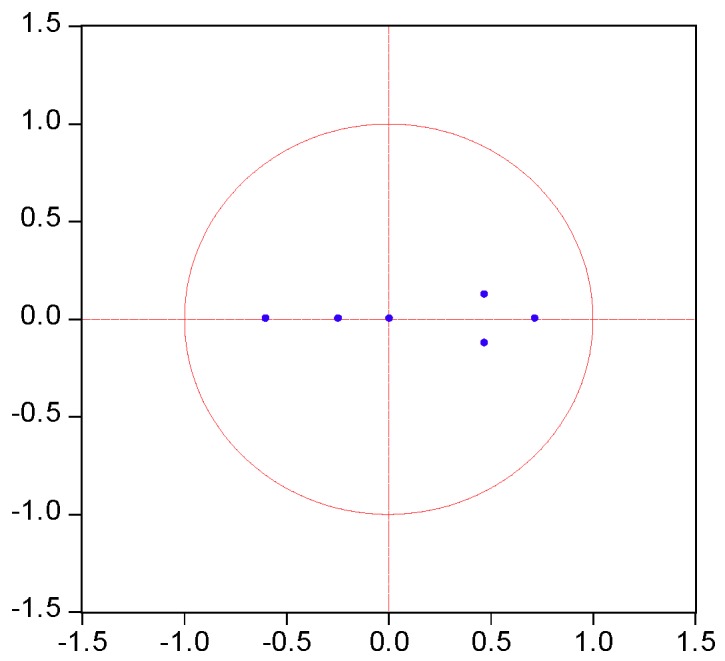
AR roots graph of the VAR model.

**Figure 5 ijerph-17-03321-f005:**
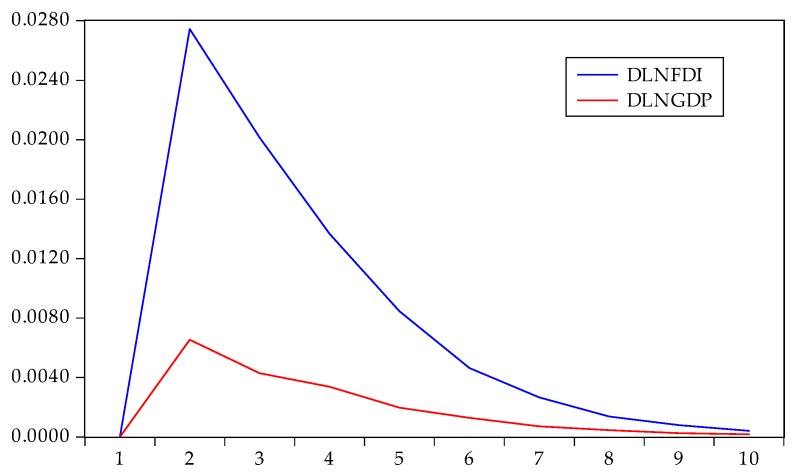
Impulse response of DLNEC to DLNFDI and DLNGDP.

**Figure 6 ijerph-17-03321-f006:**
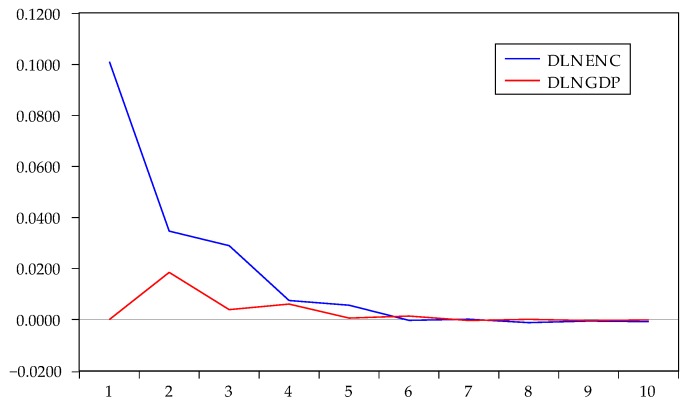
Impulse response of DLNFDI to DLNEC and DLNGDP.

**Figure 7 ijerph-17-03321-f007:**
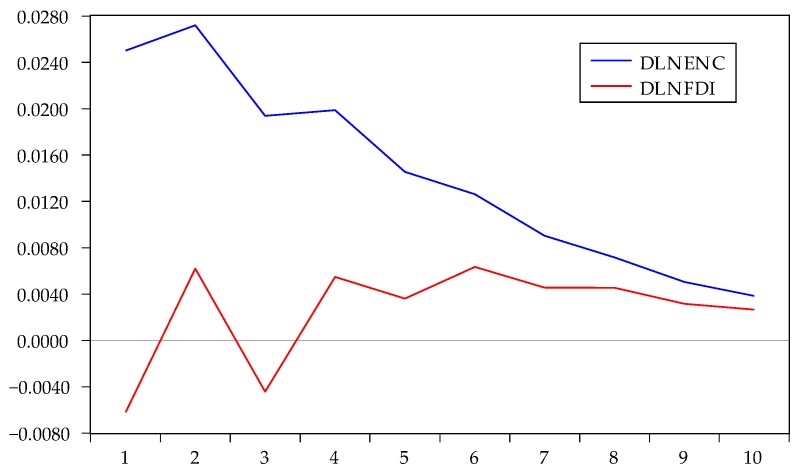
Impulse response of DLNGDP to DLNEC and DLNFDI.

**Table 1 ijerph-17-03321-t001:** Industries structure of FDI in Zhejiang Province (2008 to 2017) (%).

Year	Primary Industry	Secondary Industry	Tertiary Industry
2008	0.45	69.22	30.33
2009	0.80	64.96	34.24
2010	0.82	61.51	37.67
2011	1.54	52.17	46.28
2012	0.64	49.90	49.46
2013	0.57	43.80	55.63
2014	0.52	37.50	61.98
2015	0.53	42.41	57.06
2016	0.73	40.71	58.56
2017	0.45	34.30	65.25

Data source: Zhejiang Statistical Yearbook.

**Table 2 ijerph-17-03321-t002:** Sector structure of FDI in Zhejiang Province (2008 to 2017) (%).

Year	Secondary Industry					Tertiary Industry						
	S0	S1	S2	S3	S4	T1	T2	T3	T4	T5	T6	T7
2008	67.80	5.07	6.38	3.57	5.34	11.44	2.08	5.85	6.15	1.24	0.03	2.06
2009	63.21	4.82	6.29	3.64	5.86	14.43	5.20	2.54	5.72	1.93	0.08	2.11
2010	60.26	6.04	4.84	3.55	5.59	21.16	3.38	3.35	3.88	2.36	0.05	1.11
2011	51.28	3.37	4.13	2.31	5.06	24.81	7.42	2.72	5.80	3.02	0.16	1.38
2012	49.01	4.55	3.66	2.71	5.24	20.10	9.54	2.69	5.88	4.53	0.41	4.92
2013	42.66	4.18	4.11	4.33	4.53	24.39	7.58	4.29	6.04	4.25	1.07	6.62
2014	36.13	2.88	3.02	2.55	3.52	29.52	11.39	4.01	5.75	5.85	1.39	1.82
2015	40.94	2.29	2.78	1.07	2.87	17.91	7.49	6.58	7.90	8.17	4.84	2.06
2016	38.62	6.34	3.97	1.58	2.50	12.12	12.16	7.00	6.70	8.43	7.41	3.59
2017	30.52	3.67	3.81	1.37	2.38	10.18	13.07	11.77	10.21	11.73	3.23	3.48

Data source: Zhejiang Statistical Yearbook. Notes: the major sector i listed in the table by the labels as follows. S0: Manufacturing. S1: Chemical materials and chemical products manufacturing. S2: Communication equipment, computer and other electronic equipment manufacturing. S3: Special equipment manufacturing. S4: General equipment manufacturing. T1: Real Estate. T2: Wholesale and retail trades. T3: Information transmission, software, and information technology. T4: Leasing and business services. T5: Scientific research, technical service, and geological prospecting. T6: Financial intermediation. T7: Transport, storage, and post.

**Table 3 ijerph-17-03321-t003:** Summary statistics.

	Unit	Definition	Mean	Maximum	Minimum	Std. Dev.	Observations
LNFDI	10,000 USD	LN of FDI Inflow	12.341	13.317	11.083	0.877	25
LNGDP	100,000,000 USD	LN of Real GDP	6.653	7.864	5.295	0.893	25
LNENC	10,000 t of standard coal equivalent	LN of Energy Consumption	9.227	9.954	8.305	0.586	25

Note: LN: the natural logarithm of the variable.

**Table 4 ijerph-17-03321-t004:** Correlation matrix of variables.

	LNENC	LNFDI	LNGDP
LNENC	1.000		
LNFDI	0.984	1.000	
LNGDP	0.986	0.957	1.000

**Table 5 ijerph-17-03321-t005:** Results of the augmented Dickey-Fuller (ADF) unit root test.

Variable	Test Form	ADF Value	5% Critical Value	10% Critical Value	Results
lnENC	(C,T,1)	−1.3859	−3.6220	−3.2486	unstable
∆lnENC	(N,N,0)	−1.6513	−1.9564	−1.6085	stable
lnFDI	(C,T,2)	−2.0145	−3.6329	−3.2547	unstable
∆lnFDI	(N,N,0)	−2.1653	−1.9564	−1.6085	stable
lnGDP	(C,T,5)	−2.7409	−3.6736	−3.2774	unstable
∆lnGDP	(N,N,0)	−2.6584	−1.9564	−1.6085	stable

Notes: C, T, and K in the test form (C,T,K) separately represent the constant term, trend term, and lag order.

**Table 6 ijerph-17-03321-t006:** VAR lag order selection criteria.

Lag	LogL	LR	FPE	AIC	SC	HQ
0	19.84106	NA	4.34 × 10^−5^	−1.531005	−1.382227	−1.495958
1	124.0945	170.5965	7.63 × 10^−9^	−10.19041	−9.595294 *	−10.05022
2	137.1298	17.77545 *	5.59 × 10^−9^ *	−10.55726	−9.515806	−10.31192
3	147.3690	11.17004	5.78 × 10^−9^	−10.66991 *	−9.182125	−10.31943 *

Notes: * indicates lag order selected by the criterion. LR: sequential modified LR test statistic (each test at 5% level). FPE: Final prediction error. AIC: Akaike information criterion. SC: Schwarz information criterion. HQ: Hannan-Quinn information criterion.

**Table 7 ijerph-17-03321-t007:** Johansen cointegration test.

**Unrestricted Cointegration Rank Test (Trace)**				
Hypothesized No. of CEs	Eigenvalue	Trace Statistic	0.05 Critical Value	Prob. **
None *	0.707968	52.70580	42.91525	0.0040
At most 1	0.475967	25.62619	25.87211	0.0536
At most 2	0.404662	11.40978	12.51798	0.0760
**Unrestricted Cointegration Rank Test (Maximum Eigenvalue)**				
Hypothesized No. of CEs	Eigenvalue	Max-Eigen Statistic	0.05 Critical Value	Prob. **
None *	0.707968	27.07961	25.82321	0.0340
At most 1	0.475967	14.21641	19.38704	0.2400
At most 2	0.404662	11.40978	12.51798	0.0760

Notes: * denotes rejection of the hypothesis at 0.05 level. ** MacKinnon-Haug-Michelis (1999) *p*-values. CEs: Cointegrating Equation(s).

**Table 8 ijerph-17-03321-t008:** Vector autoregression estimates.

	DLNENC	DLNFDI	DLNGDP
DLNENC(−1)	−0.352330	−1.484423	0.174795
	(0.44860)	(1.52799)	(0.37409)
	[−0.78540]	[−0.97149]	[0.46725]
DLNENC(−2)	0.029716	−0.383719	0.361481
	(0.29020)	(0.98847)	(0.24200)
	[0.10240]	[−0.38819]	[1.49370]
DLNFDI(−1)	0.299993	0.747094	0.091096
	(0.11332)	(0.38600)	(0.09450)
	[2.64723]	[1.93550]	[0.96396]
DLNFDI(−2)	0.073945	0.395019	−0.176610
	(0.11725)	(0.39937)	(0.09778)
	[0.63066]	[0.98911]	[−1.80629]
DLNGDP(−1)	0.287366	0.813831	0.431354
	(0.29313)	(0.99843)	(0.24444)
	[0.98035]	[0.81511]	[1.76465]
DLNGDP(−2)	−0.078224	−0.361741	0.137539
	(0.10421)	(0.35495)	(0.08690)
	[−0.75065]	[−1.01913]	[1.58271]
C	0.030818	0.064315	0.016047
	(0.02478)	(0.08440)	(0.02066)
	[1.24375]	[0.76203]	[0.77660]

Notes: Standard errors in ( ) and t-statistics in [ ].

**Table 9 ijerph-17-03321-t009:** VAR stability condition check.

Root	Modulus
0.719805	0.719805
−0.599376	0.599376
0.471352 − 0.123378i	0.487232
0.471352 + 0.123378i	0.487232
−0.244291	0.244291
0.007276	0.007276

No root lies outside the unit circle. VAR satisfies the stability condition.

**Table 10 ijerph-17-03321-t010:** VAR granger causality/block exogeneity Wald tests.

Dependent Variable	Excluded	Chi-sq	df	Prob.
DLNENC	DLNFDI	7.105543	2	0.0286
DLNENC	DLNGDP	1.461576	2	0.4815
DLNFDI	DLNENC	1.010674	2	0.6033
DLNFDI	DLNGDP	1.631909	2	0.4422
DLNGDP	DLNENC	2.302331	2	0.3163
DLNGDP	DLNFDI	4.681834	2	0.0962

Notes: Chi-sq: Chi-squared test Statistics. df: degree of freedom. Prob.: Probability.

**Table 11 ijerph-17-03321-t011:** Results of variance decomposition of DLNENC, DLNFDI, and DLNGDP.

Period	Variance Decomposition of DLNENC				Variance Decomposition of DLNFDI				Variance Decomposition of DLNGDP			
	S.E.	DLNENC (%)	DLNFDI (%)	DLNGDP (%)	S.E.	DLNENC (%)	DLNFDI (%)	DLNGDP (%)	S.E.	DLNENC (%)	DLNFDI (%)	DLNGDP (%)
1	0.0412	100.00	0.0000	0.0000	0.1404	51.8247	48.1753	0.0000	0.0344	52.9402	3.2663	43.7935
2	0.0550	73.6677	24.9190	1.4132	0.1607	44.1762	54.4979	1.3259	0.0453	66.4065	3.7451	29.8485
3	0.0611	67.3145	31.0470	1.6385	0.1726	41.1362	57.6632	1.2007	0.0505	68.1469	3.7780	28.0752
4	0.0634	64.7175	33.4774	1.8051	0.1742	40.5328	58.1678	1.2994	0.0549	70.8422	4.2000	24.9578
5	0.0642	63.8209	34.3283	1.8508	0.1751	40.2338	58.4787	1.2875	0.0572	71.6730	4.2636	24.0633
6	0.0645	63.5522	34.5722	1.8756	0.1752	40.2081	58.4995	1.2925	0.0591	71.8258	5.1503	23.0239
7	0.0646	63.4681	34.6491	1.8828	0.1752	40.1924	58.5151	1.2925	0.0600	71.8107	5.5628	22.6265
8	0.0646	63.4467	34.6670	1.8863	0.1752	40.1953	58.5123	1.2924	0.0607	71.6945	6.0049	22.3006
9	0.0646	63.4403	34.6722	1.8875	0.1752	40.1956	58.5115	1.2929	0.0610	71.6259	6.2114	22.1627
10	0.0646	63.4388	34.6731	1.8881	0.1752	40.1956	58.5106	1.2929	0.0612	71.5694	6.3613	22.0693

Cholesky ordering: DLNENC, DLNFDI, DLNGDP.

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
