# Peer review of "An Empirical Analysis of Energy Consumption, FDI and High Quality Development Based on Time Series Data of Zhejiang Province"

_ijerph, 2020, doi:10.3390/ijerph17093321_

Round 1

Reviewer 1 Report

The paper is interesting and deals with an important topic of energy consumption, FDI and high quality development in China. However, there are few comments to authors.

In the Literature review, authors should add few most recent (2014-2020) research, especially regarding FDI and growth nexus.

In part 3.1.2, explain in more detail distrubution of FDI by industries. Which sector s are included in the analysis? Which sector did receive the most FDI?

In part 4.2, explain which FDI data are used? Flow or stock, total or per capita? 

Reviewer 2 Report

Authors in the literature review also should discuss halo and haven effects, it enriches the literature review. What is the novelty of this paper? Authors presenting the results did not reveal the reasons. The discussion section is missed. Authors also should mention about lag effect, which is important when FDI is analysing. 

Reviewer 3 Report

The Introduction is inadequate: it is too short and does not discuss the originality aspects of the paper.

Descriptive statistics are missed.

Additional unit root and stationairity tests results are needed.

Robustness cheks are missed.

Comparisons with previous rstudies are missed.

The editing is poor.

The paper needs a proof-reading by a native speaker.

Round 2

Reviewer 2 Report

Accept

Reviewer 3 Report

Accept